# PROGRESSIVE DATA-FREE DIFFUSION DISTILLATION

## ABSTRACT

While one-step distillation achieves strong single-step generation, these methods are not inherently flexible for multi-step sampling. Efforts to adapt them beyond one step frequently lead to a reliance on training data, poor generation quality at early intermediate steps, and significant computational demands. To overcome these limitations, we propose Progressive Multi-step Diffusion Distillation (PMDD), a unified framework that generalizes one-step distillation to the multi-step setting. PMDD adopts a recursive training strategy in which an N-step student is progressively refined into an N+1-step student with minimal finetuning. This process is enabled by a data-free sampling mechanism for generating intermediate states and an unforget loss that maintains the generation quality across steps. Together, these innovations allow PMDD to match or surpass a teacher model with only a handful of function evaluations, while providing scalable, data-free training and substantially reduced computational overhead. Extensive experiments demonstrate that our method not only outperforms established few-step diffusion approaches but also gains teacher-level-exceeded performance, with FID 1.95 on ImageNet $64 \times 64$ and FID 8.26 on zero-shot COCO $512 \times 512$, making a new state of the art in multi-step data-free distillation with significantly lower resource demands.

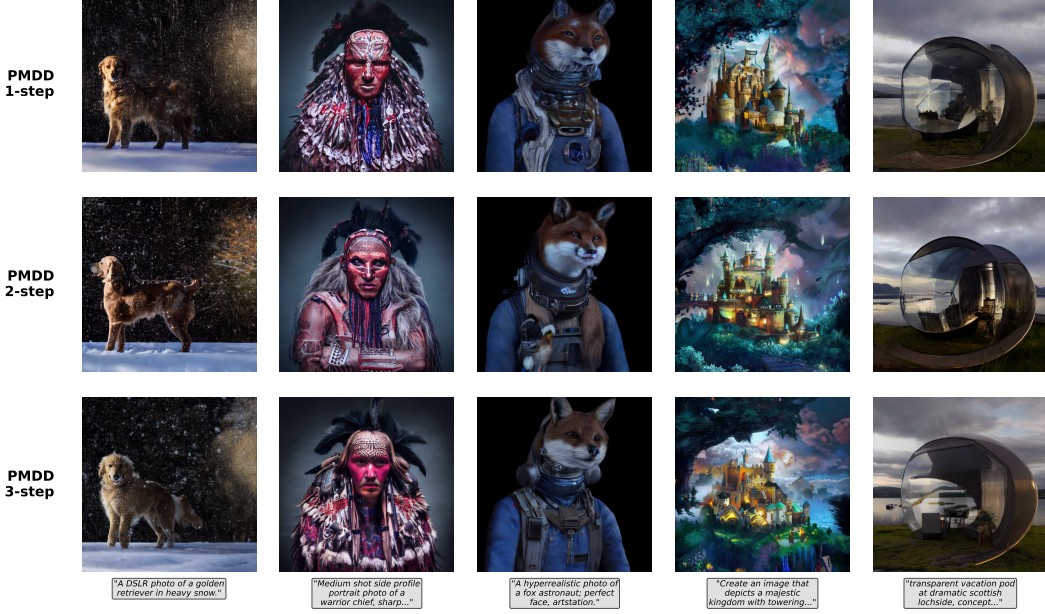

PMDD
1-step

PMDD
2-step

PMDD
3-step

"A DSLR photo of a golden retriever in heavy snow."

"Medium shot side profile portrait photo of a warrior chief, sharp..."

"A hyperrealistic photo of a fox astronaut; perfect face, artstation."

"Create an image that depicts a majestic kingdom with towering..."

"transparent vacation pod at dramatic scottish lochside, concept..."

Figure 1: $512 \times 512$ samples produced by our 3-step generator distillation of SD v1.5. All images are produced from a single unified model.

## 1 INTRODUCTION

Diffusion models have achieved remarkable success in generative tasks, demonstrating state-of-the-art performance across a wide range of domains such as image generation (Song et al., 2019; 2020b; Ho et al., 2020; Song et al., 2020a), audio synthesis (Chen et al., 2021; Kong et al., 2021), and text generation (Austin et al., 2021; Gulrajani & Hashimoto, 2023; Lou et al., 2024). Diffusion-based image generation models adopt an iterative denoising process which gradually removes noise from a noisy intermediate sample to reconstruct a high-quality image. However, the sampling process is inherently slow, typically requiring hundreds of neural function evaluations (NFEs), which makes the models expensive for many real-world applications.

To overcome this limitation, recent research has applied the distillation approach to distill a (diffusion) teacher model into a (diffusion) student model. A common strategy is to directly match the deterministic outputs of the teacher's iterative denoising process with those of the student in one or a few steps (Luhman & Luhman, 2021; Salimans & Ho, 2022; Song et al., 2023; Luo et al., 2023a; Dao et al., 2024; Kim et al., 2024); though such trajectory-matching still underperforms the teacher. In contrast, distributional-matching methods, motivated by frameworks such as GMMNs (Li et al., 2015) and GANs, bypass trajectory approximation by learning a one-step mapping from noise to clean data, ensuring that the student matches the teacher's overall output distribution (Luo et al., 2024; Nguyen & Tran, 2024; Yin et al., 2024b;a; Zhou et al., 2024). While promising, these methods have several limitations. First, they often lack the flexibility to support multi-step sampling for higher fidelity, which is critical in high-fidelity text-to-image generation where one-step is often insufficient, and multiple steps are required to refine outputs. Second, even when multi-step extensions are possible, they remain strongly dependent on data. Additionally, multistep models when sampled with only a few steps perform poorly, while still requiring substantial computational resources during the training process. ***For example, Multistep Moment Matching*** (Salimans et al., 2024) ***required 256 TPUv5e chips for two weeks of training, DMD v2 consumed 64 A100 GPUs in over a day.***

In this paper, we unify prior one-step diffusion distillation approaches under a general multi-step framework and directly address these bottlenecks. Specifically, to extend the framework beyond a single step, we introduce a progressive training strategy that incrementally expands an $N$-step teacher model into an $N + 1$-step student model, achieving improved generation fidelity with minimal finetuning overhead. However, naively applying this framework introduces two key challenges: (1) maintaining high-quality generation of intermediate latent samples, and (2) avoiding catastrophic forgetting of earlier iterations. To address these challenges, we propose a novel data-free sampling approach for intermediate states, enabling strong performance without requiring external data. To mitigate catastrophic forgetting, we introduce an unforget loss, which preserves the generation quality across iterations and substantially improves the few-step sampling setting.

We evaluate our approach across various tasks, including conditional image generation on CIFAR-10 (Krizhevsky, 2009), ImageNet 64×64 (Russakovsky et al., 2015), and zero-shot text-to-image generation on MS COCO 512×512 (Lin et al., 2014). As shown in experimental results, our one-step model consistently surpasses prior distillation methods, including Diff-Instruct (Luo et al., 2024), Distribution Matching Distillation (Yin et al., 2024b) and Consistency Models (Song et al., 2023), and even teacher models in some cases. In the multi-step setting, PMDD scales predictably with the number of steps, and outperforms other diffusion distillation methods, especially Few-step Score Identity Distillation (Zhou et al., 2025), achieving *a new state-of-the-art* in multi-step data-free distillation with **FID 8.26** on MS-COCO 2014-30k. These results are obtained with far fewer finetuning steps and substantially less computation; PMDD is trained in 5-6 days using at most **3 H100 GPUs**. In addition, compared to teacher models requiring tens or hundreds of NFEs, PMDD achieves comparable FID with 10×-20× higher efficiency.

## 2 PRELIMINARY

One-step diffusion distillation involves learning a generator $g_\phi(x_T, T)$ (typically referred to as student) that can generate data samples $x_0$ from Gaussian noise samples $x_T \sim \mathcal{N}(0, I)$ by leveraging a pretrained diffusion model (typically referred to as teacher). A standard approach to this problem is to match the data distributions $p_\phi(x_0)$ and $p_\theta(x_0)$ characterized by $g_\phi$ and the pretrained teacher,

respectively by minimizing the following KL divergence:

$$\mathcal{L}_{\text{KL}}(\phi) := D_{\text{KL}}(p_\phi(x_0) \| p_\theta(x_0)) \tag{1}$$

However, directly minimizing this KL divergence is difficult. Therefore, in practice, we minimize its variational upper bound (Ho et al., 2020; Song et al., 2020b):

$$\mathcal{L}_{\text{VUB}}(\phi) := \sum_{t=1}^{T} D_{\text{KL}}(p_\phi(x_{t-1}|x_t) \| p_\theta(x_{t-1}|x_t)) \tag{2}$$

Here, $p_\theta(x_{t-1}|x_t)$ is the parameterized backward transition distribution of the teacher while $p_\phi(x_{t-1}|x_t)$ can be regarded as the backward transition distribution of an "imaginary" diffusion model that captures $p_\phi(x_0)$. $p_\phi(x_{t-1}|x_t)$ can be parameterized in the same way as $p_\theta(x_{t-1}|x_t)$ with parameters that can be adapted from those of $p_\theta(x_{t-1}|x_t)$.

Since $p_\theta(x_{t-1}|x_t)$ is typically parameterized as $p(x_{t-1}|x_t, x_\theta(x_t, t))$ where $x_\theta(x_t, t)$ is a parametric approximation of $\mathbb{E}_{p_\theta(x_0|x_t)}[x_0]$ (Song et al., 2020a; Kingma et al., 2021), we can also express $p_\phi(x_{t-1}|x_t)$ as $p(x_{t-1}|x_t, x_\phi(x_t, t))$ with $x_\phi(x_t, t)$ approximating $\mathbb{E}_{p_\phi(x_0|x_t)}[x_0]$. Consequently, minimizing $\mathcal{L}_{\text{VUB}}(\phi)$ becomes minimizing the denoised-sample matching (DM) loss $\mathcal{L}_{\text{DM}}(\phi)$ below:

$$\mathcal{L}_{\text{DM}}(\phi) := \mathbb{E}_{x_0^T \sim g_\phi(x_T, T), t, \epsilon, x_t}\left[w_x(t) \|x_\phi(x_t, t) - x_\theta(x_t, t)\|_2^2\right] \tag{3}$$

where $x_T \sim \mathcal{N}(0, I)$, $t \sim \mathcal{U}(1, T)$, $\epsilon \sim \mathcal{N}(0, I)$, $x_t = a_t x_0^T + \sigma_t \epsilon$, and $w_x(t) > 0$ denotes the time-dependent loss coefficient w.r.t. the denoised-sample parameterization.

The main challenge when minimizing this loss is that $x_\phi(x_t, t)$ is unknown. One way to get around this problem is replacing it with the following surrogate loss (Poole et al., 2022):

$$\tilde{\mathcal{L}}_{\text{NM}}(\phi) := \mathbb{E}_{x_0^T \sim g_\phi(x_T, T), t, \epsilon, x_t}\left[w_\epsilon(t)(\epsilon_\theta(x_t, t) - \epsilon)\right]$$

where $\epsilon_\theta(x_t, t)$ can be derived from $x_\theta(x_t, t)$ and $x_t$ via Tweedie's formula (Efron, 2011). However, minimizing $\tilde{\mathcal{L}}_{\text{NM}}(\phi)$ is not equivalent to minimizing the KL divergence between $p_\phi(x_t)$ and $p_\theta(x_t)$ in Eq. 1. Consequently, this loss can lead to low-quality and low-diversity samples from the student network $g_\phi$, as observed in (Wang et al., 2024).

A better approach is to find a good approximation of $x_\phi(x_t, t)$ in Eq. 3. This can be done by training an adapted denoising network $x_\varphi$ on clean samples generated by $g_\phi$, using the following loss:

$$\mathcal{L}_{\text{adapted}}(\varphi) = \mathcal{L}_{\text{DM}}(\varphi) := \mathbb{E}_{x_0^T \sim g_\phi(x_T, T), t, \epsilon, x_t}\left[w_x(t) \|x_\varphi(x_t, t) - x_0^T\|_2^2\right] \tag{4}$$

After training $x_\varphi$, we update $g_\phi$ using a version of $\mathcal{L}_{\text{DM}}(\phi)$ with $x_\phi(x_t, t)$ replaced by $x_\varphi(x_t, t)$:

$$\mathcal{L}_{\text{DM}}(\phi) \approx \mathbb{E}_{x_0^T \sim g_\phi(x_T, T), t, \epsilon, x_t}\left[w_x(t) \|x_\varphi(x_t, t) - x_\theta(x_t, t)\|_2^2\right] \tag{5}$$

To stabilize training, prior works (Wang et al., 2024; Nguyen & Tran, 2024; Yin et al., 2024b) replace the full gradient $\nabla_\phi \mathcal{L}_{\text{NM}}(\phi)$ with a modified gradient:

$$\tilde{\nabla}_\phi \mathcal{L}_{\text{DM}}(\phi) := \mathbb{E}_{x_0^T \sim g_\phi(x_T, T), t, \epsilon, x_t}\left[w_x(t)(x_\varphi(x_t, t) - x_\theta(x_t, t))\frac{\partial g_\phi(x_T)}{\partial \phi}\right] \tag{6}$$

In practice, $\epsilon_\varphi$ and $g_\phi$ are optimized alternately by minimizing $\mathcal{L}_{\text{DM}}(\varphi)$ and updating $\phi$ with $\tilde{\nabla}_\phi \mathcal{L}_{\text{DM}}(\phi)$. More recently, Zhou et al. (Zhou et al., 2024) introduce the score identity distillation (SiD) loss into $\mathcal{L}_{\text{DM}}(\phi)$, which enables robust and stable training without the need for gradient modification. Their student loss takes the form:

$$\mathcal{L}_{\text{student}}(\phi) = \mathbb{E}_{x_0^T \sim g_\phi(x_T, T), t, \epsilon, x_t}\left[w_x(t) \|x_\varphi(x_t, t) - x_\theta(x_t, t)\|_2^2\right]$$

$$+ \alpha \mathbb{E}_{x_0^T = g_\phi(x_T, T), t, \epsilon, x_t}\left[w_x(t)(x_\theta(x_t, t) - x_\varphi(x_t, t))^\top (x_\theta(x_t, t) - x_0^T)\right]$$

$$= \mathcal{L}_{\text{DM}}(\phi) + \alpha \mathcal{L}_{\text{SiD}}(\phi)$$

## 3 METHOD

Most diffusion distillation methods either rely on the teacher model's original training data Song et al. (2023); Xie et al. (2024); Yin et al. (2024b) or are restricted to one-step distillation Gu et al. (2023); Nguyen & Tran (2024). In contrast, we study a more general and challenging setting: *data-free multistep distillation*. Our goal is to train an $n$-step student model $g_\phi$ capable of generating clean samples $x_0$ from any time steps $t_i$ ($1 \le i \le n$) under the constraint $0 < t_1 < t_2 < \ldots < t_n = T$, all without any access to clean training data.

The key difficulty lies in obtaining intermediate samples $x_{t_i} \sim p(x_{t_i})$ for $t_i < T$. With clean data, this is trivial: draw $x_0$ from the dataset and then generate $x_{t_i}$ via the forward process $p(x_{t_i}|x_0)$. In the one-step case, we can directly sample from $\mathcal{N}(0, \mathrm{I})$. Unfortunately, neither of these options applies in the data-free multistep scenario.

A naive solution is to sample $x_T \sim \mathcal{N}(0, \mathrm{I})$ and run the teacher's backward process to obtain $x_{t_i}$. Yet this simulation-based approach becomes computationally expensive as $t_i$ approaches 0, making large-scale training impractical. To overcome this, we propose a *progressive distillation* strategy, where the student $g_\phi$ is distilled in multiple stages, sequentially from $t_n$ down to $t_1$. Concretely, in the first stage ($t_n = T$), we train $g_\phi$ with Gaussian inputs using the distillation framework described in Section 2. Once $g_\phi$ can generate clean samples from step $t_n$ down to $t_{i+1}$, we further adapt it to handle step $t_i$, repeating this process until reaching $t_1$. To sample $x_{t_i}$, we begin with $x_{t_n} \sim \mathcal{N}(0, \mathrm{I})$ and recursively apply:

$$x_0^{t_k} = \mathrm{sg}\left(g_\phi\left(x_{t_k}, t_k\right)\right), \quad x_{t_{k-1}} = a_{t_{k-1}} x_0^{t_k} + \sigma_{t_{k-1}} \epsilon \tag{7}$$

where $k$ runs from $n$ to $i + 1$, $\epsilon \sim \mathcal{N}(0, \mathrm{I})$, and sg denotes the stop-gradient operator. Since each $x_0^{t_k}$ approximates samples from $p(x_0)$, the resulting $x_{t_i}$ closely follows $p(x_{t_i})$. We then pass $x_{t_i}$ through $g_\phi$(with gradients enabled) to obtain $x_0^{t_i}$ and alternately optimize $g_\phi$ and the adapted denoising network $x_\varphi$ under the following distillation objectives:

$$\mathcal{L}_{\mathrm{adapted}}^i(\varphi) = \mathbb{E}_{x_0^{t_i} = g_\phi\left(x_{t_i}, t_i\right), t, \epsilon, x_t}\left[w_x(t)\left\|x_\varphi(x_t, t) - x_0^{t_i}\right\|_2^2\right] = \mathcal{L}_{\mathrm{DM}}^i(\varphi) \tag{8}$$

$$
\begin{aligned}
\mathcal{L}_{\mathrm{student}}^i(\phi) = {}& \mathbb{E}_{x_0^{t_i} = g_\phi\left(x_{t_i}, t_i\right), t, \epsilon, x_t}\left[w_\epsilon(t)\left\|x_\varphi(x_t, t) - x_\theta(x_t, t)\right\|_2^2\right] \\
& + \alpha \mathbb{E}_{x_0^{t_i} = g_\phi\left(x_{t_i}, t_i\right), t, \epsilon, x_t}\left[w_x(t)\left(x_\theta(x_t, t) - x_\varphi(x_t, t)\right)^\top\left(x_\theta(x_t, t) - x_0^{t_i}\right)\right] \\
& + \beta \sum_{k=i+1}^n \mathbb{E}_{x_{t_k}}\left[\left\|g_\phi\left(x_{t_k}, t_k\right) - g_{\mathrm{old}}^{t_k}\left(x_{t_k}, t_k\right)\right\|_2^2\right]
\end{aligned}
\tag{9}
$$

$$= \mathcal{L}_{\mathrm{DM}}^i(\phi) + \alpha \mathcal{L}_{\mathrm{SID}}^i(\phi) + \beta \mathcal{L}_{\mathrm{unforget}}^i(\phi) \tag{10}$$

Here, $t \sim \mathcal{U}(1, T)$, $\epsilon \sim \mathcal{N}(0, \mathrm{I})$, and $x_t = a_t x_0^{t_i} + \sigma_t \epsilon$. The last term in Eq. 9 plays a critical role in preventing $g_\phi$ from catastrophically forgetting the learned multi-step mappings. In this term, $g_{\mathrm{old}}^{t_k}$ is the previous version of the student distilled at step $t_k$ up to step $t_{i+1}$.

## 4 EXPERIMENT

We assess the effectiveness of our method for distilling pretrained diffusion models on both class-conditional image generation and text-to-image generation tasks. For class-conditional generation, we adopt CIFAR-10 (Krizhevsky, 2009) and ImageNet $64 \times 64$ (Russakovsky et al., 2015) as benchmarks, using the pretrained teacher models from (Karras et al., 2022). For text-to-image generation, we distill from a pretrained Stable Diffusion v1.5 (Rombach et al., 2022) and evaluate on MS-COCO 30k (Lin et al., 2014), following standard practice in prior work (Yin et al., 2024b; Salimans et al., 2024; Zhou et al., 2025).

### 4.1 CLASS-CONDITIONAL IMAGE GENERATION

We benchmark PMDD against recent diffusion distillation methods on CIFAR-10 $32 \times 32$ and ImageNet $64 \times 64$. We follow the implementation of DMD (Yin et al., 2024b), where we generate

Table 1 (a) CIFAR-10:

| Type | Method | NFE ($\downarrow$) | FID ($\downarrow$) |
|------|--------|------|------|
| Teacher | VP-EDM (Karras et al., 2022) | 35 | 1.79 |
| One-step | GET-Base (Yin et al., 2024b) | 1 | 6.25 |
| | Meng et al. (Meng et al., 2023) | 1 | 5.98 |
| | DMD (w/o reg.) ✓ (Yin et al., 2024b) | 1 | 5.58 |
| | Diff-Instruct ✓ (Luo et al., 2024) | 1 | 4.19 |
| | DMD (w/o KL) (Yin et al., 2024b) | 1 | 3.82 |
| | DMD (Yin et al., 2024b) | 1 | 2.66 |
| | SiD ($\alpha = 1.0$) ✓ (Zhou et al., 2024) | 1 | 1.93 |
| Multi-step | Progressive Distillation$^\dagger$ (Salimans & Ho, 2022) | 1 | 9.12 |
| | | 2 | 4.51 |
| | Consistency Distillation$^\dagger$ (Song et al., 2023) | 1 | 3.55 |
| | | 2 | 2.93 |
| | CTM (Kim et al., 2024) | 1 | 1.73 |
| | | 2 | 1.63 |
| | PMDD (ours) ✓ | 1 | 2.52 |
| | | 2 | 2.19 |

(a) CIFAR-10

Table 1 (b) ImageNet $64 \times 64$:

| Type | Method | NFE ($\downarrow$) | FID ($\downarrow$) |
|------|--------|------|------|
| Teacher | VP-EDM (Karras et al., 2022) | 79 | 2.64 |
| One-step | BOOT ✓ (Gu et al., 2023) | 1 | 16.3 |
| | DFNO (Zheng et al., 2023a) | 1 | 7.83 |
| | TRACT (Berthelot et al., 2023) | 1 | 7.43 |
| | SwiftBrush ✓ (Nguyen & Tran, 2024) | 1 | 5.85 |
| | Diff-Instruct ✓ (Luo et al., 2024) | 1 | 5.57 |
| | DMD (Yin et al., 2024b) | 1 | 2.62 |
| | DMD v2 (w/o GAN) ✓ (Yin et al., 2024a) | 1 | 2.61 |
| | DMD v2 (Yin et al., 2024a) | 1 | 1.28 |
| | SiD ($\alpha = 1.0$) ✓ (Zhou et al., 2024) | 1 | 2.02 |
| Multi-step | Progressive Distillation (Salimans & Ho, 2022) | 1 | 15.39 |
| | | 2 | 8.95 |
| | Consistency Distillation (Song et al., 2023) | 1 | 6.20 |
| | | 2 | 4.70 |
| | Moment Matching (Salimans et al., 2024) | 1 | 3.0 |
| | | 2 | 3.86 |
| | CTM (Kim et al., 2024) | 1 | 1.92 |
| | | 2 | 1.73 |
| | PMDD (ours) ✓ | 1 | 2.60 |
| | | 2 | 1.95 |

(b) ImageNet $64 \times 64$

Table 1: Results on CIFAR-10 (left) and ImageNet $64 \times 64$ (right) of our method and baselines. Data-free distillation methods are marked with ✓, unconditional methods are marked with $^\dagger$.

50,000 images for every 1000 training iterations in order to calculate the FID metric (Heusel et al., 2017), and report the best model achieving the lowest FID during evaluation. At each stage, the student and adapted networks are reinitialized from their best-performing checkpoints. We summarize the results in Table 1.

**One-step** PMDD achieves an NFE-1 FID of 2.52 on CIFAR-10 and 2.60 on ImageNet $64 \times 64$, outperforming prior state-of-the-art data-dependent one-step distillation methods such as TRACT, DFNO, and DMD, as well as recent data-free approaches including SwiftBrush and Diff-Instruct. PMDD ranks only behind Score Identity Distillation (SiD) and CTM; however, CTM is data-dependent, while SiD is substantially more computationally demanding (see discussion below). Compared to pretrained teacher diffusion models such as DDIM, PMDD achieves $\approx 3.3\times$ lower FID while being $10\times$ faster.

**Multistep** PMDD surpasses established data-dependent multi-step methods, including Progressive Distillation, Consistency Distillation and Multistep Moment Matching, achieving an FID of 1.95 with only two function evaluations (NFE = 2). It also outperforms SiD (FID = 2.02); however, SiD requires the equivalent of 1 billion synthetic training images (121k iterations with a very large batch size), whereas our method achieves competitive performance using only around 34M synthetic images in total - nearly 30× fewer. Furthermore, compared to its pretrained teacher model, PMDD observed $\approx 1.25\times$ lower FID while being $\approx 40\times$ faster and more efficient.

### 4.2 TEXT-TO-IMAGE GENERATION

To assess the scalability of our approach to large-scale dataset, we distill a latent-space model at $512 \times 512$ resolution using Stable Diffusion v1.5 (Rombach et al., 2022) following prior work settings (Yin et al., 2024a). Evaluation is conducted on zero-shot MS COCO, where we report both FID and CLIP score to measure fidelity and text-image alignment.

Table 2 shows that our 3-step PMDD surpasses nearly all diffusion distillation methods, with the only exception of Moment Matching — a data-dependent approach that requires up to 8 NFEs for sampling and massive compute (256 TPUv5 cores for over two weeks of training). In contrast, PMDD achieves an FID of 8.50 with only 3 sampling steps, a data-free method trained in 8 days on 3 H100 GPUs. Remarkably, PMDD even outperforms its teacher model (SDv1.5, 50 NFEs, 8.52 FID). It also outperforms Few-step Score Identity Distillation, a concurrent data-free distillation method, while requiring fewer sampling steps and yielding better FID. These results establish PMDD as the new state of the art in few-step data-free distillation.

**Behavior of PMDD under varying inference budgets** Table 2 reports the best model trained for each step count. We further analyze the robustness of PMDD when generating images with varying inference budgets under a single unified model. Table 3 shows that in both 2-step and 3-step settings, the unforget loss $\mathcal{L}_{\text{unforget}}^i(\phi)$ yields stronger final-step performance than competing methods, and

| Method | NFE ($\downarrow$) | COCO FID$_{30k}$($\downarrow$) | CLIP Score ($\uparrow$) |
|---|---|---|---|
| **Base Models** | | | |
| SD v1.5 (CFG = 3) (Rombach et al., 2022) | 512 | 8.78 | - |
| SD v1.5 (CFG = 8) (Rombach et al., 2022) | 512 | 13.45 | 0.322 |
| **Diffusion Distillation (One-step)** | | | |
| DMD (CFG=3) (Yin et al., 2024b) | 1 | 11.49 | - |
| DMD (CFG=8) (Yin et al., 2024b) | 1 | | 0.32 |
| SwiftBrush (Nguyen & Tran, 2024) | 1 | 16.67 | 0.29 |
| SwiftBrush+PG+NASA (Nguyen et al., 2024) | 1 | 9.94 | 0.31 |
| InstaFlow-1.7B (Liu et al., 2023) | 1 | 11.8 | 0.309 |
| DMDv2 (CFG = 1.75) (Yin et al., 2024a) | 1 | 8.35 | 0.30 |
| **Diffusion Distillation (Multistep)** | | | |
| LCM-LoRA (Luo et al., 2023b) | 4 | 23.62 | |
| PeRFlow (Yan et al., 2024) | 4 | 18.59 | - |
| SLAM (Xu et al., 2024) | 4 | 10.06 | |
| Moment Matching (CFG = 0) (Salimans et al., 2024) | 8 | 7.25 | |
| DMDv2 w/o GAN (CFG = 1.75) ($\checkmark$) (Yin et al., 2024a) | 1 | 9.35 | 0.304 |
| (reimplemented) | 2 | 10.44 | 0.301 |
| | 3 | 9.18 | 0.303 |
| Few-step Score Identity Distillation (Zero-CFG) ($\checkmark$) (Zhou et al., 2025) | 1 | 9.63 | 0.321 |
| | 2 | 8.75 | 0.315 |
| | 4 | 8.52 | 0.308 |
| PMDD (CFG = 1.75) ($\checkmark$) | 1 | 10.41 | 0.302 |
| PMDD (CFG = 1.75) ($\checkmark$) | 2 | 8.63 | 0.30 |
| PMDD (CFG = 1.75) ($\checkmark$) | 3 | 8.50 | 0.302 |
| PMDD (CFG = 1.0) ($\checkmark$) | 3 | **8.26** | 0.298 |

Table 2: Comparison of image generation methods on 30k COCO-2014 prompts, following a standard evaluation protocol. Methods that are data-free ($\checkmark$)

| Method | NFE=3 | NFE=2 | NFE=1 |
|---|---|---|---|
| Guided Distill. | - | 33.25 | 108.21 |
| LCM | - | 13.31 | 35.36 |
| Self-corrected Flow Distillation | - | 11.46 | 11.91 |
| DMDv2 w/o GAN (reimplemented) | - | 10.44 | 16.22 |
| PMDD (CFG = 1.75) | - | **8.63** | **11.67** |
| DMDv2 w/o GAN (reimplemented) | 9.18 | 10.36 | 23.32 |
| PMDD (CFG = 1.75) | **8.50** | 10.07 | **12.65** |

Table 3: FID comparison of diffusion distillation methods under varying sampling budgets

| Unforget Weight ($\beta = 1.0$) | External Sampling of $x_{t_i}$ | $\mathcal{L}_{\text{SiD}}$ | CIFAR-10 | ImageNet 64 $\times$ 64 |
|---|---|---|---|---|
| | | | 5.89 | 8.01 |
| $\checkmark$ | | | 3.02 | 3.71 |
| $\checkmark$ | $\checkmark$ | | 2.94 | 3.58 |
| $\checkmark$ | $\checkmark$ | $\checkmark$ | 2.21 | 1.99 |

Table 4: Ablation Study of 2-step model on CIFAR-10 and ImageNet 64 $\times$ 64. FID is reported for all experiments.

maintains high fidelity even at low step counts. We examine the role of $\mathcal{L}_{\text{unforget}}^i (\phi)$ more closely in Section 4.3.

Figure 1 demonstrates that, conditioned on the same initial noise $x_T$, PMDD consistently preserves a coherent global image structure across different sampling steps. Subsequent steps typically refine fine details, such as facial expressions, while the overall structure remains intact. This shows the possibility of utilizing a single model across all steps, suitable for varying inference budgets depending on available resources and desired generation quality.

## 4.3 ABLATION STUDIES

We conduct extensive ablation studies on our distilled model, which explores the impact of three key factors: the unforget weight ($\beta = 1.0$), the inclusion of additional score identity loss $\mathcal{L}_{\text{SiD}}^i (\phi)$, and the role of external sampling of $x_{t_i}$ during training and sampling using previously trained models. Table 4 demonstrates that model performance is mainly driven by two key components: the score identity loss $\mathcal{L}_{\text{SiD}}^i (\phi)$ and the unforget loss $\mathcal{L}_{\text{unforget}}^i (\phi)$.

Table 5 indicates that under $\mathcal{L}_{\text{DM}}^i (\phi)$, PMDD's performance shows consistent improvements as the number of sampling steps increases. In contrast, while $\mathcal{L}_{\text{SiD}}^{(t_i)} (\phi)$ yields strong results under 2-step inference, it does not scale effectively to additional steps, limiting further improvements in image quality. Moreover, when applied to higher-dimensional image generation tasks such as Stable Diffusion, $\mathcal{L}_{\text{SiD}}^i(\phi)$ leads to poor performance and fails to learn successfully. Extending $\mathcal{L}_{\text{SiD}}^{(t_i)} (\phi)$ to large-scale text-to-image generation task for PMDD is left for future work.

| | CIFAR-10 | | ImageNet $64 \times 64$ | |
|---|---|---|---|---|
| Inference Steps | $\mathcal{L}_{\mathrm{DM}}^i(\phi)$ | $\mathcal{L}(\phi) + \alpha\mathcal{L}_{\mathrm{SiD}}^i(\phi)$ | $\mathcal{L}_{\mathrm{DM}}^i(\phi)$ | $\mathcal{L}(\phi) + \alpha\mathcal{L}_{\mathrm{SiD}}^i(\phi)$ |
| FID (NFE = 1) | 3.49 | 2.52 | 3.70 | 2.60 |
| FID (NFE = 2) | 2.91 | 2.19 | 3.58 | 1.95 |
| FID (NFE = 3) | 2.86 | 2.48 | 3.47 | 2.14 |

Table 5: Ablation of the loss term on distilling a 2-step model on CIFAR-10 and ImageNet $64 \times 64$. By default, we use our best hyper-parameters $\alpha = 1.0$ and $\beta = 0.3$.

Figure 2 further explores the impact of varying the unforget loss weight $\beta$ on CIFAR-10 and ImageNet $64 \times 64$. The effect is minimal for 2-step sampling but becomes significant in learning to unforget 1-step. For Stable Diffusion, Table 6 and Figure 3 indicate that performance is highly sensitive to this weight, with optimal results achieved when $\alpha \in [0.01, 0.1]$, highlighting the critical role of precise loss balancing in our framework. Larger weights overemphasize the unforget objective at the expense of distribution matching loss, preserving fidelity in earlier steps while degrading final-step quality.

Table 6 compares the effect of external sampling of $x_{t_i}$. In 2-step sampling, the difference between using and not using external sampling is marginal; however, in 3-step sampling the effect is substantial (Figure 3). With CFG = 1.75, training without external sampling (brown line) requires roughly twice as many iterations to match the convergence speed of training with external sampling (red line). This occurs because, without external sampling, the model must learn to map from a constantly changing $x_{t_i}$ (generated by the current model and therefore not fixed), while also handling unforget at earlier steps. In contrast, external sampling fixes $x_{t_i}$, allowing the model to focus on reducing the FID of the final step, while requiring a larger unforget weight to preserve fidelity at earlier steps.

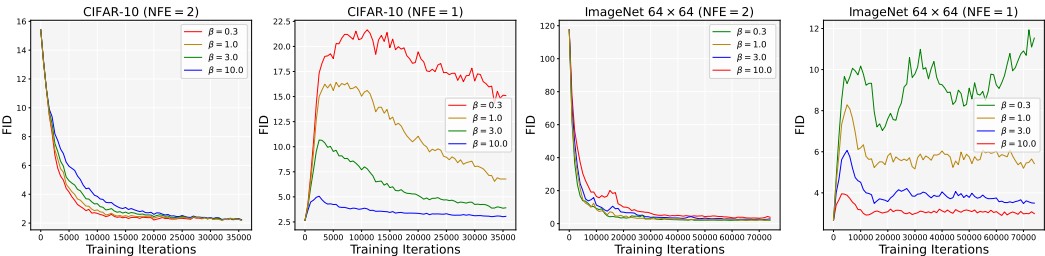

Figure 2: Effect of unforget loss $\beta$ on 1-step while training 2-step for CIFAR-10 and ImageNet $64 \times 64$ ($\alpha = 1.0$)

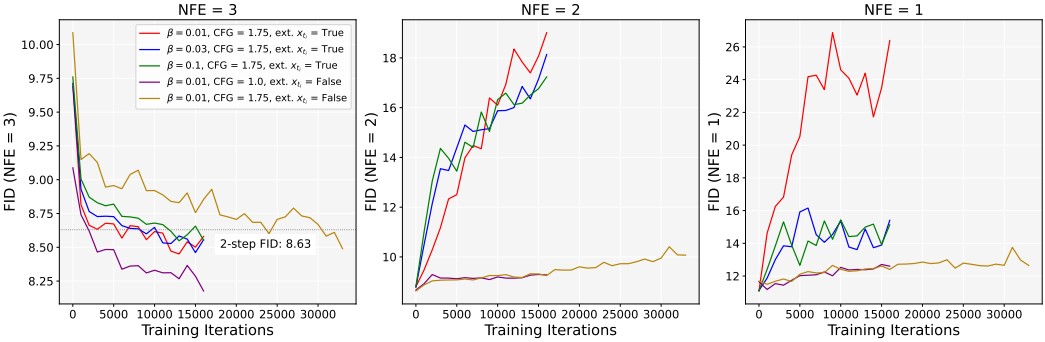

Figure 3: Effect of unforget loss $\beta$ on 3-step inference for COCO 2014

## 5 RELATED WORK

Training-free methods employ higher-order numerical solvers to expedite the backward process, especially high-order SDE Solvers. For instance, Stochastic Explicit Exponential Derivative-free

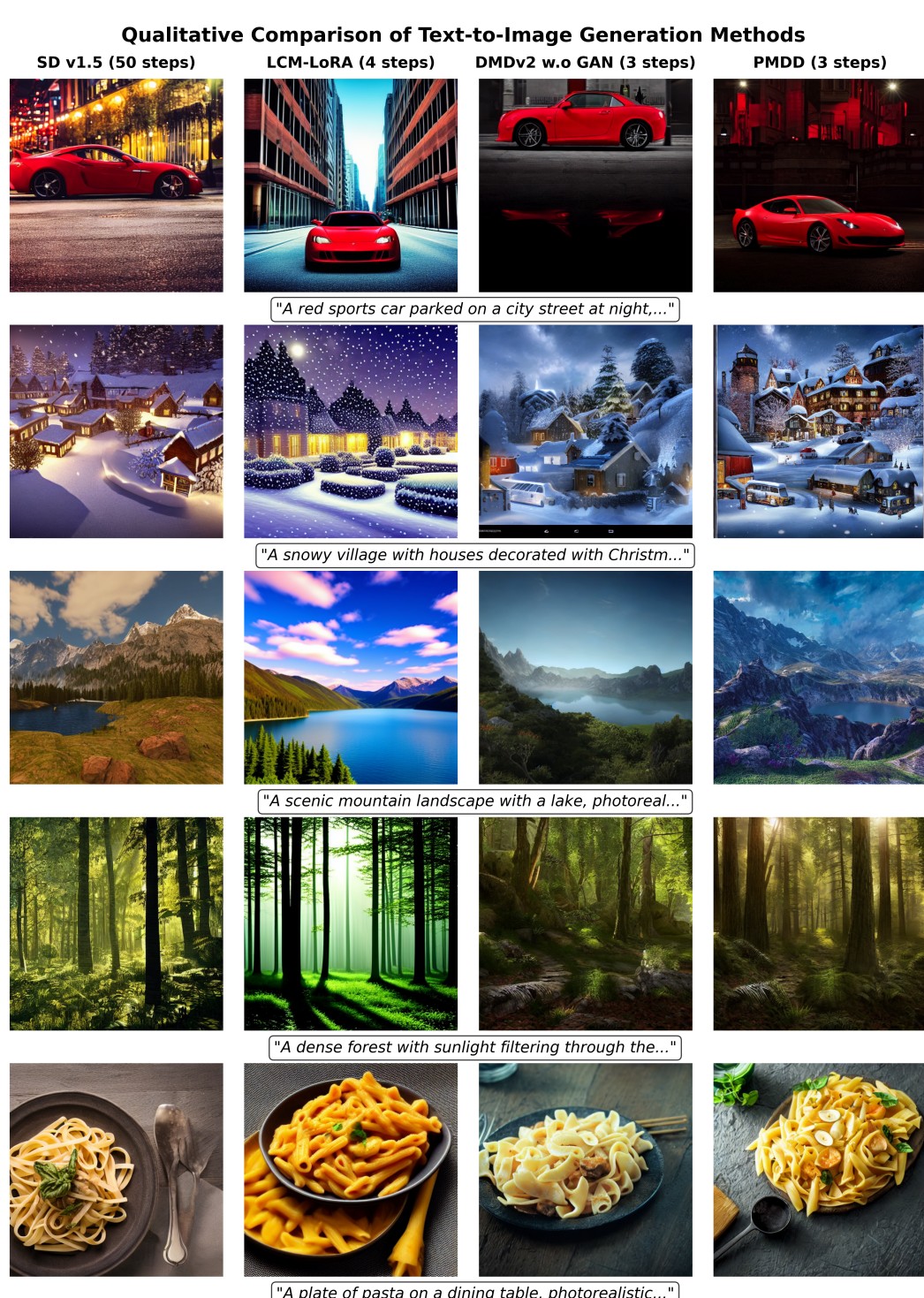

Figure 4: Comparison of text-to-image generation across Stable Diffusion v1.5 (50 steps) and other multistep diffusion distillation methods such as LCM-LoRA, PMDD, and DMD v2. Our model (final column) attains superior quality compared to other methods, with comparable or faster inference speed.

| | Ext. Sampling × | | Ext. Sampling ✓ | |
|---|---|---|---|---|
| Unforget Weight | FID (2 steps) | FID (1 step) | FID (2 steps) | FID (1 step) |
| $\lambda = 0.3$ | 8.63 | 11.67 | 8.57 | 11.81 |
| $\lambda = 1.0$ | 8.79 | 11.42 | 8.66 | 11.52 |
| $\lambda = 3.0$ | 8.89 | 10.98 | 8.61 | 10.50 |
| $\lambda = 10.0$ | 9.44 | 10.07 | 9.00 | 9.99 |

Table 6: Ablation of unforget weight and external sampling of $x_{t_i}$ on 2-step inference for COCO 2014 trained in 16K iterations.

Solvers (SEEDS)(Gonzalez et al., 2024) employs an exponential time-differencing approach separating linear terms for analytical evaluation, while SA-Solver (Xue et al., 2024) applies Adams-Bashforth integrator which controls noise injection via hyper-parameter $\tau$. In general, diffusion samplers utilizing enhanced SDE solvers tend to be slower than those based on high-order ODE solvers (Lu et al., 2022a;b; Zheng et al., 2023b), reasoned by ODE's deterministic nature simplifying the denoising process. High-order ODE solvers typically exploit the special structures of the diffusion generation process. (Liu et al., 2022) designs the VP ODE semi-linear structure, while (Zhang & Chen, 2022; Lu et al., 2022a) further expand this concept and utilize an exponential integrator method to simplify the process. Notably, UniPC (Zhao et al., 2024), which integrates a corrector into DPM-Solver++ Lu et al. (2022b), unifies various existing methods under a predictor-corrector framework.

An alternative approach focuses on aligning the distributions of the student and teacher across different time steps. SwiftBrush (Nguyen & Tran, 2024) adapts 3D distribution matching techniques from Score Distillation Sampling (Poole et al., 2023) and Variational Score Distillation (Wang et al., 2024) to 2D image synthesis by replacing the 3D NeRF rendering component with a 2D text-to-image generator. Yin et al. (2024b) further leverages this framework by incorporating an extra regression loss for better generation capabilities. Zhou et al. (2024) generalizes this idea by replacing the reverse KL-Divergence used in original work with Fisher Divergence, featuring DMD as its special case and achieving a more general framework for student-teacher distribution alignment. A concurrent work - Zhou et al. (2025) leverages this framework to extend to multistep data-free sampling by jointly training $N$ steps simultaneously with a single adapted network $x_\varphi(x_t, t)$ to approximate $g_\phi(x_{t_i}, t_i)$ where $x_t = a_t g_\phi(x_{t_i}, t_i) + \sigma_t \epsilon$ for all $t_i$.

Through extensive experiments against DMDv2 (Yin et al., 2024a) and (Zhou et al., 2025), we find that relying on a single adapted network is insufficient. In contrast, our method introduces a progressive training mechanism, employing a separate adapted network $x_\varphi(x_t, t)$ for each $t_i$. This strategy along with the unforget loss $\mathcal{L}_{\text{SiD}}(\phi)$ achieves superior performance compared to both Yin et al. (2024a) and Zhou et al. (2025).

# 6 CONCLUSION

In conclusion, our progressive multi-step diffusion distillation framework effectively overcomes the limitations of prior one-step and distributional-matching approaches, achieving high-fidelity generation with significantly reduced computational cost. By introducing data-free intermediate sampling and an unforget loss, our method preserves generation quality across iterations and enables efficient few-step sampling. Experimental results demonstrate that PMDD consistently outperforms existing distillation methods and even teacher models in some cases, setting a new state-of-the-art in multi-step data-free diffusion distillation while requiring far fewer resources.

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
