# A   IMPLEMENTATION DETAILS

This section presents an overview of our implementation details across CIFAR-10, ImageNet $64 \times 64$, and zero-shot COCO 2014. We closely follow the implementation of DMD and DMDv2 throughout our experiments, and open-source our training and evaluation code for reproducibility.

## A.1   CIFAR-10

**Stage 1 (One-step distillation)**

We distill a one-step student model from the EDM Pretrained model Karras et al. (2022), using edm-cifar10-32x32-cond-vp for class-conditional image generation. Training is performed with AdamW optimizer (learning rate of 1e-5, a weight decay 0.01, and beta parameters $(0.9, 0.999)$), a total batch size of 256 on 4 A100 GPUs for 200K iterations. Following the original implementation, we adopt a two-time-update-rule: the adapted network is updated 5 times per a student update. The score identity distillation loss is implemented based on Score Identity Distillation Zhou et al. (2024) with default hyper-parameters: $\alpha = 1.2$ and loss_scaling_generator $= 100$.

**Stage 2 and 3 (Progressive multi-step training)**

In the subsequent training stages, we initialize the student model with the best FID checkpoint from the previous stage, and finetune with AdamW optimizer. For stage 2, we use a learning rate of 5e-6, a weight decay of 0.01, beta parameters $(0.9, 0.999)$, batch size 256, and train for 35K iterations with 4 A100 GPUs. For stage 3, we reduce the learning rate to 8e-7 and train for 65K iterations under the same setup. We also adopt a progressive training time-split schedule $[[0, 1], [0, 0.5], [0, 0.25]]$, where in the final stage the student learns the mapping from $x_{0.25}$ to $x_0$. We also evaluate an equal-width schedule $[[0, 1], [0, 0.67], [0, 0.33]]$, and observe similar results. Further exploration of scheduling strategies is left for future work. Our ablational studies show that different unforget weights are marginal effect; therefore, we decide to use an unforget weight of 1.0.

## A.2   IMAGENET $64 \times 64$

**Stage 1 (One-step distillation)**

We distill a one-step student model from the EDM Pretrained model Karras et al. (2022), using edm-imagenet-64x64-cond-adm for class-conditional image generation. Training is performed with AdamW optimizer (learning rate of 2e-6, a weight decay 0.01, and beta parameters $(0.9, 0.999)$), a total batch size of 72 on 4 A100 GPUs for 400K iterations. Following the original implementation, we adopt a two-time-update-rule: the adapted network is updated 5 times per a student update. The score identity distillation loss is implemented based on Score Identity Distillation with default hyper-parameters: $\alpha = 1.2$ and loss_scaling_generator $= 100$. .

**Stage 2 and 3 (Progressive multi-step training)**

Subsequent stages progressively extend the distilled student model to support multi-step sampling. Each stage is initialized from the best FID checkpoint of the previous stage and finetuned with AdamW.

- Stage 2: We use a learning rate of 8e-7, a weight decay of 0.01, beta parameters $(0.9, 0.999)$, batch size 72, unforget weight 0.3, and train for 40K iterations with 4 A100 GPUs, which takes approximately 1 day.

- Stage 3: We adopt the same configuration to train the 3-step PMDD model.

## A.3   SD V1.5

**Stage 1 (One-step distillation)**

We distill a one-step student model from the SD v1.5 model Rombach et al. (2022) using prompts from LAION-Aesthetic 6.25+ dataset. Unlike DMD v2, our approach does not require collecting images for the GAN discriminator. Training is performed with AdamW optimizer (learning rate of 5e-6, a weight decay 0.01, and beta parameters $(0.9, 0.999)$), a total batch size of 120 on 3

H100 GPUs for 120K iterations. Following prior work, we adopt a two-time-update-rule where the adapted network is updated 10 times per student update. We also implemented the score identity distillation loss and found that it did not yield meaningful improvements during training.

**Stage 2 and 3 (Progressive multi-step training)**

Subsequent stages progressively extend the distilled student model to support multi-step sampling. Each stage is initialized from the best FID checkpoint of the previous stage and finetuned with AdamW.

- Stage 2: We use a learning rate of 8e-7, a weight decay of 0.01, beta parameters $(0.9, 0.999)$, batch size 20, unforget weight 0.3, and train for 16K iterations with 1 H100 GPU, which takes approximately 12 hours. To accelerate convergence, we reduce the adapted network updates to 5 per student update.

- Stage 3: Using the same configuration, we train the 3-step PMDD model. Based on ablation findings that high unforget weights overemphasize preservation at the expense of distribution matching, we lower the unforget weight to 0.01 and continue training for another 16K iterations.

## B EVALUATION DETAILS

For the CIFAR-10 and ImageNet $64 \times 64$ datasets, we generate 50,000 images every 1000 iterations, and calculate the FID metric based on EDM's evaluation code Karras et al. (2022). For the zero-shot COCO 2014 text-to-image generation, we closely follow the evaluation process as reported in DMD Yin et al. (2024b), in which we generate 30,000 images using random prompts from the MS-COCO2014 validation set. These images are then downsampled to $256 \times 256$, and evaluated against roughly 40,000 real images from the validation set to calculate the FID metric. We also use the OpenCLIP-G backbone to report the CLIP score.

## C PROMPTS FOR FIGURES

We adopt DMD's paper and use the following prompts to generate the figures:

- A DSLR photo of a golden retriever in heavy snow.

- Medium shot side profile portrait photo of a warrior chief, sharp facial features, with tribal panther makeup in blue and red, looking away, serious but clear eyes, 50mm portrait, photography, hard rim lighting photography.

- A hyperrealistic photo of a fox astronaut; perfect face, artstation.

- Create an image that depicts a majestic kingdom with towering castles.

- Transparent vacation pod at dramatic scottish lochside, concept prototype, ultra clear plastic material, editorial style photograph.

- A red sports car parked on a city street at night, photorealistic, sharp focus, HDR, cinematic lighting, ultra-detailed reflections.

- A snowy village with houses decorated with Christmas lights, photorealistic, cinematic lighting, high resolution, ultra-detailed textures.

- A scenic mountain landscape with a lake, photorealistic, ultra-detailed, cinematic lighting, high resolution.

- A dense forest with sunlight filtering through the trees, photorealistic, cinematic composition, sharp details, ultra-detailed textures.

- A plate of pasta on a dining table, photorealistic, food photography style, sharp focus, studio lighting, ultra-detailed textures.

## D ADDITIONAL QUALITATIVE SAMPLES

810
811
812
813
814
815
816
817
818
819
820
821
822
823
824
825
826
827
828
829
830
831
832
833
834
835
836
837
838
839
840
841
842
843
844
845
846
847
848
849
850
851
852
853
854
855
856
857
858
859
860
861
862
863

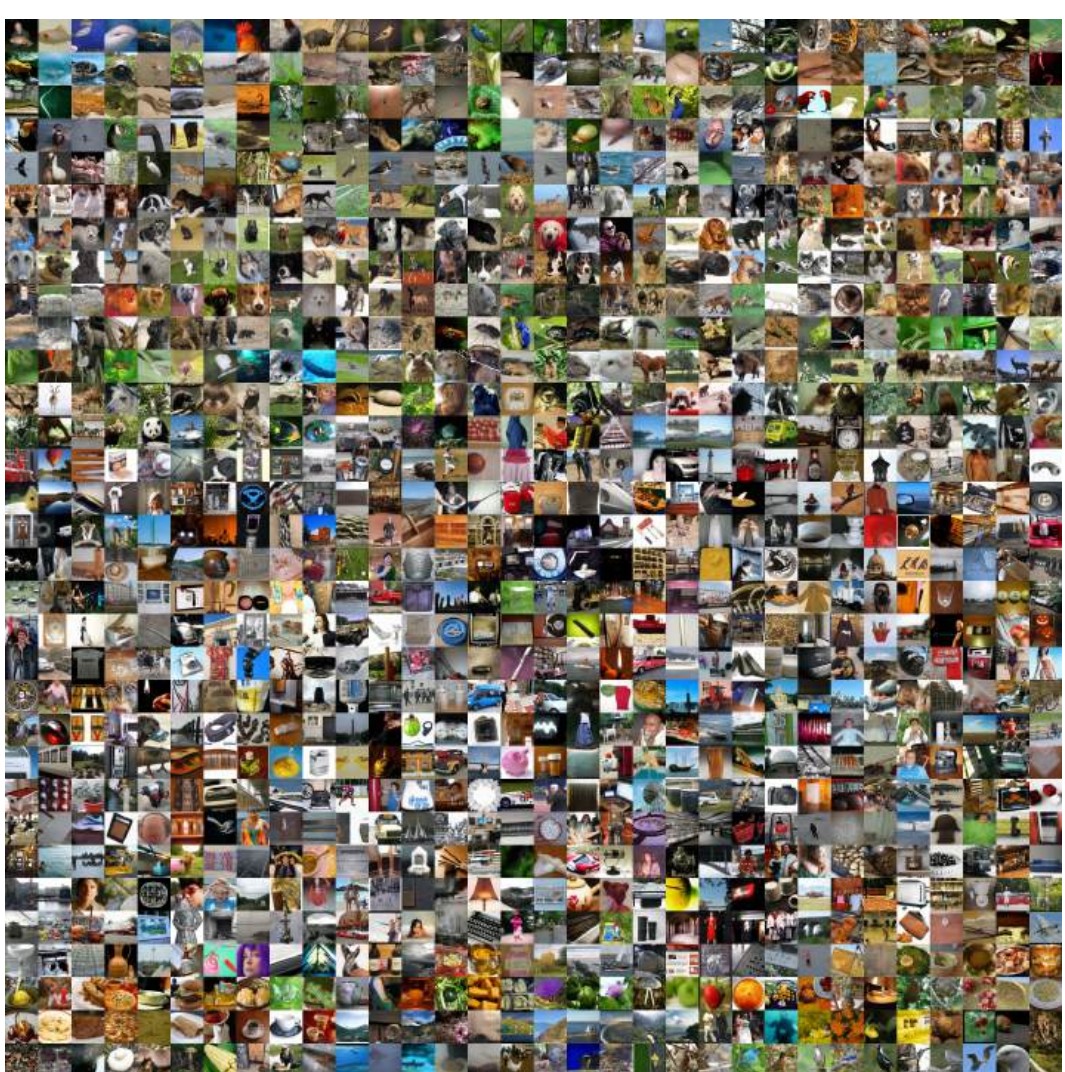

Figure 5: Samples from our 1-step student generator on ImageNet (FID=2.60).

864
865
866
867
868
869
870
871
872
873
874
875
876
877
878
879
880
881
882
883
884
885
886
887
888
889
890
891
892
893
894
895
896
897
898
899
900
901
902
903
904
905
906
907
908
909
910
911
912
913
914
915
916
917

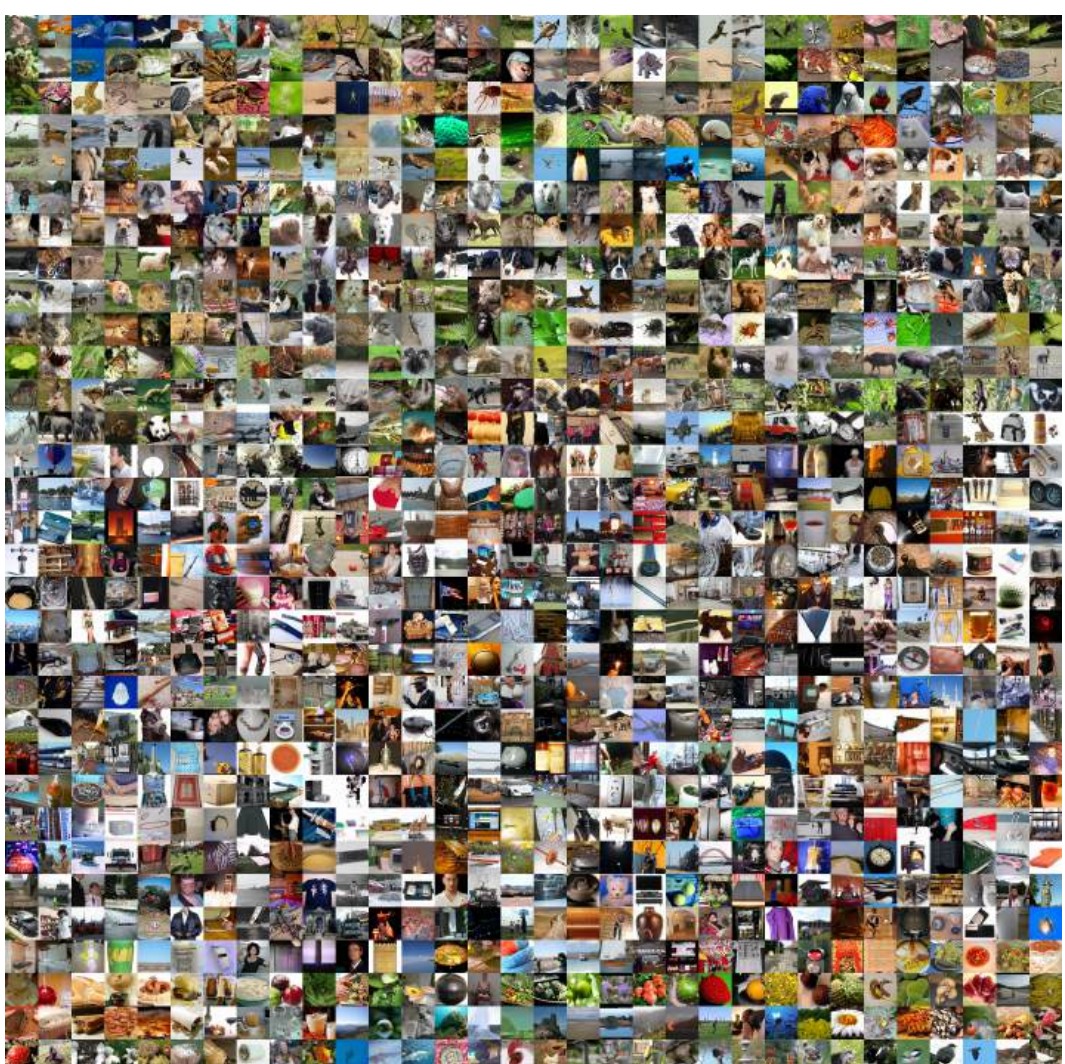

Figure 6: Samples from our 2-step student generator on ImageNet (FID=1.95).

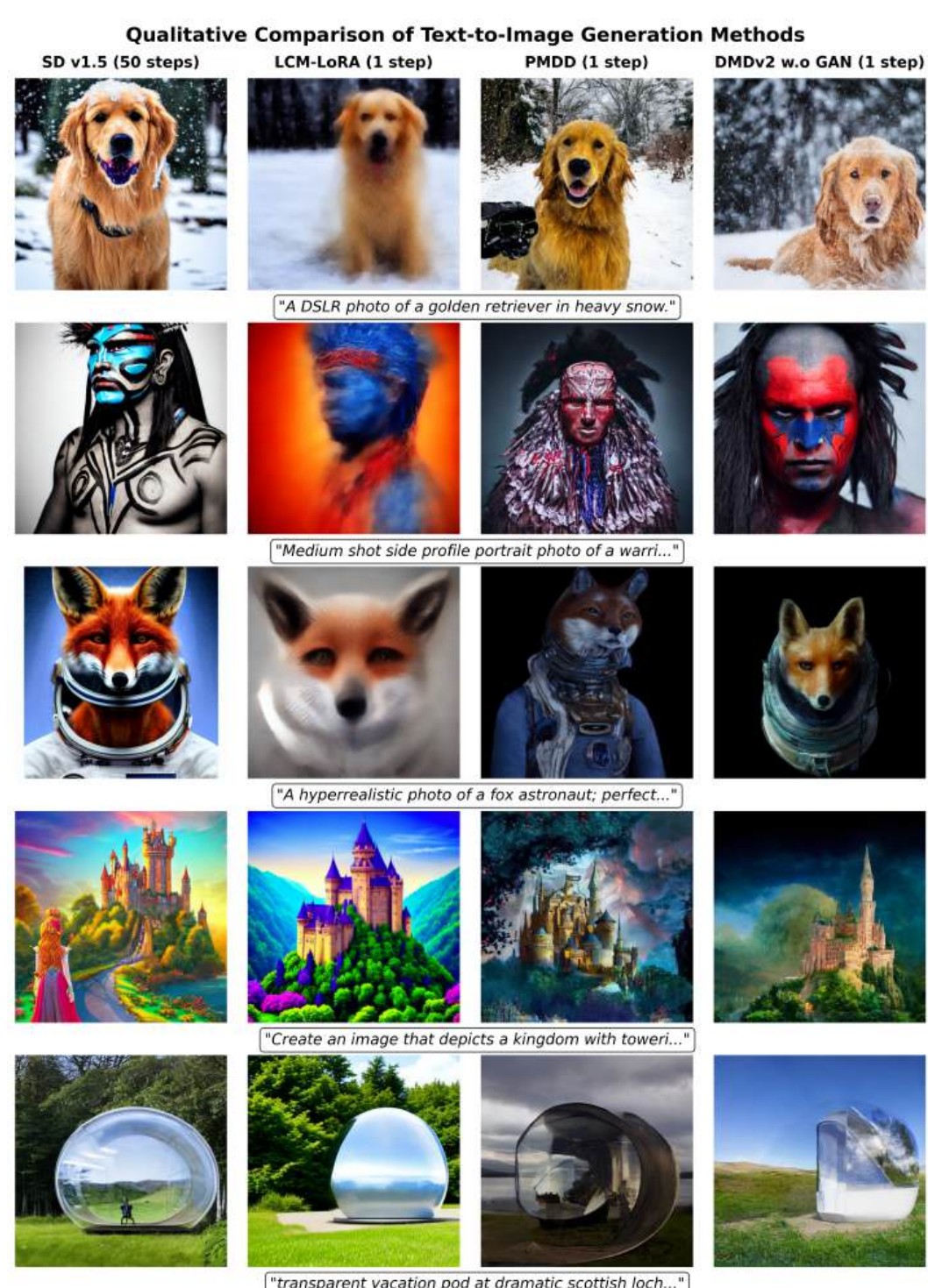

Figure 7: Comparison of text-to-image generation across Stable Diffusion v1.5 (50 steps) and one-step diffusion distillation methods such as LCM-LoRA, PMDD, and DMD v2.