# OpenReview forum: "Progressive Multistep Data-free Diffusion Distillation"
_ICLR.cc/2026/Conference — ICLR 2026 Conference Withdrawn Submission_

### Official Review · Reviewer_Aau1 · 2025-10-30

**Soundness:** 3
**Presentation:** 2
**Contribution:** 2
**Rating:** 4
**Confidence:** 4

**Summary:**

This paper introduces Progressive Multi-step Diffusion Distillation (PMDD), a framework that generalizes one-step diffusion distillation into a scalable multi-step setting. The method employs a recursive training strategy combined with data-free sampling and an “unforget loss”, aiming to achieve high-fidelity generation with reduced computation. Extensive experiments on CIFAR-10, ImageNet 64×64, and zero-shot COCO demonstrate competitive or superior performance to previous approaches.

**Strengths:**

Experimental results are comprehensive, covering both class-conditional and text-to-image settings, and demonstrate the scalability of the method.

**Weaknesses:**

**Readability issue**: The font size in Tables 1–4 is too small to read clearly in the PDF version. This significantly affects the readability of experimental results.

**Methodological clarity**: Many key equations in Section 2 and Section 3 are cited directly from prior work (e.g., Eqs. (2)–(6)) without adequate derivation or explanation. This makes the methodology section difficult to follow, especially for readers not deeply familiar with diffusion-based distillation.

Some variable definitions (e.g., $x_{\phi}$, $x_{\varphi}$) are reused across different contexts, leading to minor ambiguity.

The figures could better illustrate the recursive training process (e.g., adding a schematic overview of the multi-step progression).

The introduction and related work sections could be better organized to highlight the novelty relative to prior structured diffusion approaches.

**Questions:**

Could the authors provide at least one complete derivation example (e.g., Eq. (3) or Eq. (5)) to help readers understand the adaptation of the denoised-sample matching loss?



Please consider increasing the font size of tables and reformatting them for clarity.

---

### Official Review · Reviewer_rox6 · 2025-10-31

**Soundness:** 2
**Presentation:** 2
**Contribution:** 2
**Rating:** 4
**Confidence:** 4

**Summary:**

This paper introduces an innovative Progressive Multi-step Diffusion Distillation (PMDD) method, which builds on the concepts of PD and DMD distillation. PMDD addresses the computational overhead and lack of flexibility in traditional single-step and distribution-matching methods for generating high-quality images. By introducing data-free intermediate sampling and the unforget loss function, PMDD effectively improves generation quality and reduces computational costs. Experimental results show that PMDD outperforms existing distillation methods on multiple tasks, setting a new state-of-the-art for multi-step data-free diffusion distillation.

**Strengths:**

PMDD proposes a progressive multi-step diffusion distillation framework that overcomes the limitations of traditional single-step and distribution-matching approaches. It introduces a novel combination of data-free intermediate sampling and the unforget loss. Compared to existing distillation methods, PMDD can perform flexible and efficient few-step sampling without using real data, while maintaining high-quality generation.

**Weaknesses:**

1. PMDD is only evaluated on SDv1.5, lacking validation of its generalizability across broader frameworks, such as DiT-based models like PixArt, or downstream tasks such as video generation compression.
2. The paper discusses the impact of external sampling on performance, especially in the 3-step sampling scenario. However, how to further optimize the model without external sampling—particularly in inference stages, where external sampling can be computationally expensive—warrants further discussion.
3. The description of the implementation of PMDD in Section 3 ("Method") is somewhat simplistic, making it challenging for readers to fully understand the specific steps of the method.

**Questions:**

1. In Figure 3, for NFE=2 and NFE=1, why does the FID increase with more training iterations, contrary to the trend shown in Figure 2?
2. In the experiments, the unforget loss weight ($\beta$) significantly impacts model performance, especially at different step counts. The authors mention that larger weights may overly emphasize the unforget objective, affecting generation quality. How can the optimal $\beta$ value be automatically selected in practical applications to ensure the best performance across various tasks? Is there an adaptive method for adjusting this parameter?
3. The experiments mainly focus on low-resolution image generation, but how does PMDD perform on higher-resolution images (e.g., 1K or 2K)? Can the training maintain the same performance, or is the training difficulty higher for such high-resolution tasks?

---

### Official Review · Reviewer_SQeA · 2025-10-31

**Soundness:** 2
**Presentation:** 2
**Contribution:** 1
**Rating:** 2
**Confidence:** 4

**Summary:**

This paper proposes a multi-step distillation method called Progressive multistep diffusion distillation (PMDD) that adapts distribution matching distillation-based methods for multi-step sampling. The distillation starts by applying Score identity distillation (SiD) to get a one-step student model. This student is then progressively distilled to generate in 2 steps, 3 steps and so on. In the paper, this time-split schedule is [0, 1], [0, 5] and [0, 0.25]. The distillation loss involves the original SiD loss and an additional “unforget loss”. This loss is a L2 regularization term which ensures that the student being distilled in the next step doesn’t forget the predictions of the previous step. The paper demonstrates results on ImageNet-64, CIFAR-10 and COCO 2014 30K datasets.

**Strengths:**

1. Reduced computation effort: This method needs much fewer computational resources and training time to fine-tune one-step student to multi-step generator. As mentioned in the paper, PMDD can be trained in 5-6 days on 3 H100 GPUs.
2. The method is data-free as well as simple to understand and implement.

**Weaknesses:**

1. This method seems to require lot of hyper parameter tuning and engineering effort. Each stage of distillation needs a different set of hyper parameters and training time as mentioned in the Appendix A. Overall, the central idea is incremental from SiD. DMD-v2 also does multi-step sampling for fixed time-steps and is data free.
2. The sampling is not adaptive but rather fixed to the choice of time-split schedule. Further, the gains from multi-step sampling don't seem impressive beyond 2 steps.
3. Missing implementation details: Details such as the choice of weighting function are important in distillation but the paper does not go into the details.

**Questions:**

1. In Eq 9. in the paper, what is $w_\epsilon(t)$? Is this weighting term exclusively a function of noise $\epsilon$?
2. The paper would benefit if an algorithmic box for training is included in the appendix.
3. After the final distillation, does the same model have the ability to sample in 1, 2 and 3 steps? The time-split schedule indicates [0, 1], [0, 5] and [0, 0.25]. How does 3 step sampling work in this case where we need to go from 1 -> 0.5 -> 0.25 -> 0 as [0.5, 0.25] is missing from the time-split schedule?

---

### Official Review · Reviewer_G1ZF · 2025-11-01

**Soundness:** 2
**Presentation:** 3
**Contribution:** 2
**Rating:** 4
**Confidence:** 5

**Summary:**

This paper proposes a unified framework that extends one-step distillation to multi-step distillation, using a recursive training strategy to maintain generation quality across intermediate steps. Experimental results demonstrate that the student model even surpasses the teacher model in performance.

**Strengths:**

The paper presents an iterative approach that extends one-step distillation to multi-step distillation, allowing flexible control over the sampling steps to balance generation speed and quality.

**Weaknesses:**

1. The method section is overly simplified and fails to clearly explain the proposed method’s details. For instance, in Equation (9), the meaning, role, and computation of each term are not properly clarified.
2. The proposed method appears quite similar to existing approaches, such as the Consistency Model or the Consistency Trajectory Model, particularly in its ability to train a student model capable of generating clean samples from any time step, which is similar to the consistency constraint. The authors should discuss the differences in detail to highlight their contribution.
3. I have concerns regarding the experimental results. Since the student model is distilled without any access to clean data and is solely guided by the teacher model, theoretically, the student model’s performance should approach the teacher, not exceed it. Therefore, I would like to know how the student model to outperform the teacher on ImageNet dataset.
4. The baseline models used in the experiments are not up to date; more recent models such as HybridSD and SenseFlow should be included for comparison.
5. The proposed method seems to have low training efficiency, as it requires multiple rounds of iterative training.
6. The experiments only present results for 1-, 2-, and 3-step generations. It remains unclear whether further improvements can be achieved with more steps.
7. In the text-to-image generation experiments, comparisons with state-of-the-art models such as Flux, SD 3.5, and SDXL are missing. To my knowledge, some versions of these models can also achieve few-step generation.

**Questions:**

Please see weakness.

---

### Note · Authors · 2025-11-14

I have read and agree with the venue's withdrawal policy on behalf of myself and my co-authors.